# Knowledge, perceptions and practices on antibiotic use among Sri Lankan doctors

**Gihan Shu**[1☯¤], **Kaushika Jayawardena**[1☯¤], **Dinesh Jayaweera Patabandige**[2], **Asanka Tennegedara**[1], **Veranja Liyanapathirana**[1]*

1 Department of Microbiology, Faculty of Medicine, University of Peradeniya, Kandy, Sri Lanka, 2 Teaching Hospital, Peradeniya, Sri Lanka

☯ These authors contributed equally to this work.
¤ Current address: Teaching Hospital, Peradeniya, Sri Lanka
* veranjacl@pdn.ac.lk, veranjacl@yahoo.com

**Data Availability Statement:** All relevant data are within the paper and its Supporting information files.

## Abstract

### Introduction

Prescribers have a major role in preventing antimicrobial resistance (AMR) through appropriate prescribing. However, in countries like Sri Lanka, where continuous professional development is not mandatory for license renewal and antimicrobial stewardship is not implemented, prescribing practices go largely unchecked.

### Objectives

To identify the knowledge on antibiotic use and practices related to antibiotic prescribing among Sri Lankan doctors.

### Methods

This cross-sectional study was conducted in 2020. We used a validated, pretested Google-form based questionnaire with multiple choices, single best answer questions, polar questions (Yes/No) and five-point Likert scale questions. The Google-sheet generated was used for data analysis. Knowledge and practice scores were calculated.

### Results

Of the 262 respondents, 40.1% were males. Majority (61.8%) were aged 25–35-years and in medical practice for 0–5 years (48.9%) while 46.2% had or were engaged in post graduate studies. Knowledge scores ranged from 98.31% to 46.55% [mean:71.27% (SD±10.83); median:71.18% (IQR 64.4–79.7)]. Most (98.09%) obtained ≥50 marks while 45.8% scored more than the mean. The practice scores ranged from 100% to 0% [mean:65.33% (SD ±18.16), median:66.67% (IQR53.3–80)]. The majority (81.3%) scored ≥50 in the practice score while 52.3% achieved more than the mean practice score. The knowledge score and the practice score differed significantly (p<0.001, related sample Wilcoxon Signed Rank Test) but the knowledge and practice scores were significantly correlated [Spearman correlation, p<0.001, r = 0.343 (Bias corrected 95% CI 0.237–0.448)]. Knowledge scores and the practice scores were significantly higher in those with or undergoing postgraduate training.

**Funding:** The author(s) received no specific funding for this work.

**Competing interests:** VL has received funding from Pfizer for a non-related study. Others declare no competing interests. This does not alter our adherence to PLOS ONE policies on sharing data and materials.

## Conclusions

While the knowledge and practice scores were high, and knowledge and practice scores were correlated, the practices score was lower than that of knowledge indicating the need to encourage correct practices through means other than solely promoting knowledge.

## Introduction

Antimicrobial resistance (AMR) has been identified as one of the ten threats to global health in 2019 by the World Health Organization (WHO) [1]. The rapid emergence of resistant micro-organisms is a consequence of inappropriate use of antibiotics in many sectors including health sector [2] and has a negative impact on patient outcomes [3].

Inappropriate use of antibiotics in healthcare includes incorrect dosing, inadequacy of duration, inappropriate frequency, lack of compliance, unsupervised self-medication and over-the-counter (OTC) sale of antibiotics without prescriptions. These global issues are escalating at an alarming rate [4–7].

Although inappropriate use of antibiotics is prevalent globally, developing countries are most affected due to higher infection rates and limited resources [7]. In some countries, one of the most important factors affecting antimicrobial resistance is the lack of proper training of prescribers in proper antimicrobial use and infectious disease diagnosis. Demand and pressure from the patients have been identified as another cause driving the irrational use of antibiotics in low and middle-income countries [8]. Many studies conducted in developing countries have revealed that antibiotics are commonly used in day-to-day practice for being fever and respiratory symptoms, irrespective of etiology [8, 9]. Sri Lanka being a lower-middle-income economy is no different from other similar countries. High rates of antibiotic use at the first contact is a well-recognized issue in Sri Lanka [10]. High patient volume and fear of bacterial superinfections have been cited as reasons for antibiotic overuse in Sri Lanka [11].

Doctors are the main prescriber in many countries including Sri Lanka. In addition to the high workload and lack of facilities limiting the opportunities to deliver evidence-based care, [2] a prescriber's decision may also be influenced by factors such as updated knowledge [12], fear of losing patients' and lack of information on rational antibiotic use [13].

Community-acquired antibiotic resistance is on the rise [14] and up-to-date knowledge and correct practices regarding rational use of antibiotics among prescribers are now more important than ever. Prescribers should also consider AMR as an important factor in prescribing [15, 16]. The WHO World Health Assembly adopted an action plan to combat AMR and preserve the efficacy and availability of antimicrobials in 2015. Two of its objectives are improving awareness and understanding on AMR and optimizing the use of antimicrobials [17].

Hence this study was conducted to identify the knowledge and practices related to antibiotic use among Sri Lankan doctors across different specialties and levels of experience, in the context of AMR.

## Materials and methods

This was a cross-sectional study conducted recruiting 262 participants from various fields of Medicine working in Sri Lanka, through convenience sampling. The ethical clearance was obtained from the Ethics Review Committee, Faculty of Medicine, University of Peradeniya,

Sri Lanka (2020/EC/36) and participants indicated consent for participation through pressing the next button as informed through instructions. The questionnaire was Google form-based. It was validated by a Specialist in Medical Education and two Consultant Microbiologists with experience in similar studies and relevant clinical practice. The tool was piloted to a group of 10 medical officers and modified as needed. The finalized tool was disseminated to doctors through hospital based social media groups. Any doctor practicing in Sri Lanka, registered with the Sri Lanka Medical Council were eligible to answer the questions. One attempt was allowed per-email address; therefore, a single response was taken per-participant. The questionnaire was open to accept answers from 15/10/2021 to 12/01/2021.

The questionnaire collected socio-demographic details of the participants including their duration of practice, work unit, their current role, and their post-graduate qualification status. The questionnaire consisted multiple choice questions, single best answer questions, polar questions (Yes/No), five-point Likert scale questions, as well as open-ended questions.

The questionnaire was used to assess the knowledge on antibiotics and antimicrobial resistance using a score. For this purpose, eight defined (Yes/No/ Don't know) questions and six multiple choices were allocated. For the former group, 1 mark was awarded for the correct answer, and none were awarded to any answer labeled as incorrect/ don't know/ blank. In the multiple-choice questions, 0.25 marks was given for each correct choice in identifying beta-lactam antibiotics and causes for AMR. For the same questions, 0.25 was deducted for each incorrect choice. Negative marks were not carried forwards from question to question. The answers provided for the multiple-choice questions that assessed the knowledge on selecting the most appropriate antibiotic for resistant phenotypes [Methicillin resistant *Staphylococcus aureus* (MRSA), Carbapenem resistant Enterobacteriaceae (CRE), Extended Spectrum Beta Lactamase producing Enterobacteriaceae (ESBLE) and vancomycin resistant enterococci (VRE)] were categorized as the "correct choice", "correct choice + others", and "incorrect choice". Only the correct choice was awarded 1 one and incorrect choices were not given negative marks.

The answers provided for the questions which assessed the knowledge on abbreviations of resistant bacterial phenotypes, were initially categorized as know the meaning, don't know the meaning, and unanswered. Those that categorized as "don't know the meaning", were subcategorized as "don't know the meaning, but have heard the term" and "don't know the meaning and not heard the term" (S1 Table).

In the questions that assessed prescribing related personal experiences and perspectives, the categories "agree" and "strongly agree" were merged to a common category as "agree" and the categories "disagree" and "strongly disagree" were merged as "disagree".

Ten questions were allotted to assess the practices regarding antibiotic use (S2 Table). They were five-point Likert scale (almost always/ often/ sometimes/ rarely/ never) based or polar questions (Yes/ No). For assigning marks, "almost always" and "often" and, "rarely" and "never" were grouped together. Depending on the nature of the question, the most appropriate category was given 2 marks whereas the least appropriate was awarded 0. If the participant had stated his/her answer as "sometimes" 1 mark was given. Any question that was left unanswered was not given marks. The polar questions were awarded 1 and 0 for the correct and incorrect answers respectively.

For the question on how the dose of antibiotics was decided, 2 marks were given for "by referring to BNF or another formulary", 1 mark for any other choice except for "by remembering", and 0 was given for, "by remembering".

The answers provided for the open-ended questions were thoroughly assessed and categorized under common themes. Those that were unable to do so, were located under the theme "others".

Any questions left blank were labeled as either "blank" or "unanswered".

The questionnaire had a voluntary option to indicate if they wanted to have a feedback on the answers and the answers were emailed to those who indicated so.

The maximum mark achievable for knowledge score was 14.75 while it differed for the practice score depending on if participants were engaged in hospital practice and/or private practice. All total marks were converted to %, so the maximum achievable mark was 100% while the minimum was 0%.

Knowledge and practice scores were checked for normalcy. T test and ANOVA were used to compare the knowledge scores across groups as the knowledge score was normally distributed while the Mann-Whitney U test and Kruskal Wallis test was used to compare the practice scores as this was not normally distributed. Spearman correlation co-efficient was calculated between the knowledge and practice scores. The knowledge and practice scores for individuals were compared using the related sample Wilcoxon Signed Rank Test. Data analysis was done with SPSS (IBM statistics, version 23).

## Results

Out of 262 participants, 105 (40.1%) were male. A majority (n = 162, 61.8%) were from 25–35-year group and have been in medical practice between 0–5 years (n = 128, 48.9%), while a larger proportion (n = 141, 53.8%) were not postgraduate trained (Table 1).

A higher number of participants provided the correct answer for questions that assessed their knowledge of antibiotics (Table 2). The responses were somewhat divided upon asking antibiotics are used only to treat bacterial infections, in which 159 (60.7%) had answered "yes" while 103 (39.3%) had answered "no". Out of the latter group, 48 (46.6%) had mentioned fungal and parasitic infections, 25 (24.3%) had highlighted prophylactic use, 4 (3.9%) as gastroparesis, 3 (2.9%) as hepatic encephalopathy, 3 (2.9%) as viral infections and 12 (11.7%) had highlighted other instances such as an anti-inflammatory and immunomodulation (Table 3).

Upon asking to pick beta-lactams out of the choices provided (Table 3), 93 participants (35.5%) selected at least 1 beta-lactam correctly but only 44 (16.8%) selected all four (penicillin, cefotaxime, meropenem, and aztreonam). Although many knew that penicillin is a beta-lactam (n = 245, 93.5%) a lower number of participants were aware that aztreonam was a beta-lactam (n = 57, 21.8%). A considerable number of participants (n = 52, 19.8%) had incorrectly picked vancomycin as a beta-lactam. Two participants (0.8%) did not selected any option and 50 (19.2%) selected an incorrect antibiotic and only 4 (1.5%) had selected all 2 incorrect antibiotics (vancomycin and ciprofloxacin) coupled with other choices. Hundred and twelve (42.7%) had selected a beta-lactam(s) with an incorrect antibiotic(s).

A high number of participants had selected appropriate causes for antimicrobial resistance (Table 3). But it is noteworthy that only 94 (35.9%) and 142 (54.2%) had selected poor handwashing practices and poor infection control practices in hospitals as possible contributors.

Most participants were aware of MRSA (n = 160, 61.1%) while out of 101 participants who did not know the meaning, 100 participants (99%) had heard the abbreviation while only 1 (1%) had not heard the term either. A similar pattern was observed with regards to ESBLEs, although the group of participants who had neither heard nor knew the meaning is comparatively greater than that of MRSA (n = 49, 35%). It is noteworthy that only a few participants were aware of the meaning of CRE (n = 38, 14.5%) and the number of participants who had neither heard nor knew the meaning was significantly high (n = 157, 81.8%). A similar pattern was seen in VRE (Table 4).

Based on the responses for the questions to assess the knowledge on drug options for various resistant bacterial phenotypes, it is apparent that most participants (n = 169, 64.5%) had

**Table 1. Socio-demographic details of the participants.**

| | | n | % |
|---|---|---|---|
| **Age** | 25–35 years | 162 | 61.8 |
| | 36–45 years | 76 | 29 |
| | 46–55 years | 22 | 8.4 |
| | 56–65 years | 2 | 0.8 |
| **Sex** | Male | 105 | 40.1 |
| | Female | 157 | 59.9 |
| **Years in practice** | 0–5 | 128 | 48.9 |
| | 6–10 | 63 | 24 |
| | >10 | 70 | 26.7 |
| | Unanswered | 1 | 0.4 |
| **Post graduate qualifications** | Yes, currently enrolled | 69 | 26.3 |
| | Yes, completed | 52 | 19.8 |
| | No | 141 | 53.8 |
| **What post graduate studies** | M.D. | 83 | 68.6 |
| | M.Sc. | 15 | 12.4 |
| | *P.G. Diploma | 18 | 14.9 |
| | Other | 4 | 3.3 |
| | Unanswered | 1 | 0.8 |
| **Current role** | Academic | 2 | 0.8 |
| | Consultant | 23 | 8.8 |
| | Senior Registrar | 18 | 6.9 |
| | Registrar | 40 | 15.3 |
| | Senior House Officer | 19 | 7.3 |
| | Medical Officer | 132 | 50.4 |
| | Relief House Officer | 6 | 2.3 |
| | House Officer | 14 | 5.3 |
| | *PG trainee | 3 | 1.1 |
| | Medical Officer of Health | 5 | 1.9 |
| **Work Unit** | *ET/PCU | 13 | 5.0 |
| | Full time general practice | 1 | 0.4 |
| | Medical unit | 35 | 13.4 |
| | *Obs and Gyn unit | 13 | 5.0 |
| | *OPD | 19 | 7.3 |
| | Paediatric | 15 | 5.7 |
| | Surgical Unit | 38 | 14.5 |
| | Public Health | 17 | 6.5 |
| | Other specialties | 111 | 42.4 |
| **Engaged in active clinical service** | Yes | 237 | 90.5 |
| | No | 25 | 9.5 |
| **Type of hospital working, if engaged in active clinical service** | National Hospital | 63 | 26.6 |
| | Teaching Hospital | 63 | 26.6 |
| | Provincial General Hospital | 8 | 3.4 |
| | District General Hospital | 46 | 19.4 |
| | Divisional Hospital | 13 | 5.5 |
| | Base Hospital | 32 | 13.5 |
| | *PMCU | 4 | 1.7 |

(*Continued*)

**Table 1.** (Continued)

| | | n | % |
|---|---|---|---|
| | Specialized hospitals | 3 | 1.3 |
| | Central clinics | 1 | 0.4 |
| | National programmes | 1 | 0.4 |
| | *MOH | 1 | 0.4 |
| | Others | 1 | 0.4 |
| | Unanswered | 1 | 0.4 |
| **Engaged in private practice** | No | 179 | 68.3 |
| | Yes | 83 | 31.7 |
| **Nature of private practice** | Affiliated to private hospitals | 11 | 13.3 |
| | Independent general practice (part time or full time) | 50 | 60.2 |
| | Specialist practice | 22 | 26.5 |

*PG = Post Graduate, ET/PCU = Emergency Treatment/ Primary Care Unit, Obs and Gyn = Obstetrics and Gynaecology, OPD = Outpatient Department, PMCU = Primary Medical Care Unit, MOH = Medical Officer of Health

selected the correct choice for MRSA while 47 (17.9%) had selected the correct choice coupled with other choices, 36 (13.7%) had selected an incorrect choice and 10 (3.8%) had refrained from providing an answer. A similar pattern is observed with regards to ESBL although 74 participants (28.3%) had selected an incorrect choice. A striking observation was made with regards to CRE where only 32 participants (12.2%) had selected the correct choice and 58 (22.1%) and 166 (63.4%) had selected the incorrect choice and refrained from answering respectively. A similar pattern is seen with VRE as well (Table 5).

Table 6 depicts understanding and perspectives regarding antibiotic use. These questions were of agree/ disagree type. Eighty-one participants (30.9%) had agreed with the statement that they find it hard to select the correct antibiotic for a patient with a bacterial infection, while 121 (46.2%) had disagreed. Further,116 (44.3%) participants agreed to the fact that

**Table 2. Summary of responses related to knowledge on antibiotic use and resistance.**

| | Yes | | No | | Don't know | | Unanswered | |
|---|---|---|---|---|---|---|---|---|
| | n | % | n | % | n | % | n | % |
| **Antibiotics used to treat bacterial infections only**[*] | 159 | 60.7 | 103 | 39.3 | - | - | - | - |
| **Antibiotics can prevent development of bacterial infections** | 223 | 85.1 | 36 | 13.7 | 3 | 1.1 | - | - |
| **Administration of antibiotics within one hour to a patient with sepsis is life-saving** | 234 | 89.3 | 12 | 4.6 | 16 | 6.1 | - | - |
| **Antibiotics should be prescribed to any patient with fever** | 1 | 0.4 | 261 | 99.6 | - | - | - | - |
| **Empirical antibiotics should be escalated or de-escalated based on microbiological culture results** | 248 | 94.7 | 12 | 4.6 | 1 | 0.4 | 1 | 0.4 |
| **Age and body weight are important in deciding antibiotic use** | 259 | 98.9 | 2 | 0.8 | 1 | 0.4 | - | - |
| **Any antibiotic can be used to treat any bacterial infection** | 4 | 105 | 258 | 98.5 | - | - | - | - |
| **Combinations are always better than monotherapy** | 49 | 18.7 | 193 | 73.7 | 19 | 7.3 | 1 | 0.4 |
| **All bacterial infections can be treated with the same duration of antibiotics** | - | - | 262 | 100 | - | - | - | - |
| **Antimicrobial resistance is a world-wide problem**[*] | 261 | 99.6 | - | - | 1 | 0.4 | - | - |
| **Antimicrobial resistance is a problem in my country**[*] | 258 | 98.5 | 1 | 0.4 | 2 | 0.8 | 1 | 0.4 |

*Not considered for the knowledge score

**Table 3. Summary of responses for other antimicrobial uses, knowledge on beta lactams and causes for AMR.**

| | | n | % |
|---|---|---|---|
| **Other uses of antibiotics** | Fungal and parasitic infections | 48 | 46.6 |
| | Gastroparesis | 4 | 3.9 |
| | Hepatic encephalopathy | 3 | 2.9 |
| | Prophylaxis | 25 | 24.3 |
| | Viral infections | 3 | 2.9 |
| | Other | 12 | 11.7 |
| | Unanswered | 8 | 7.8 |
| **Selecting beta lactams** | Penicillin correctly identified | 245 | 93.5 |
| | Cefotaxime correctly identified | 152 | 58.0 |
| | Meropenem correctly identified | 101 | 38.5 |
| | Aztreonam correctly identified | 57 | 21.8 |
| | Vancomycin incorrectly identified | 52 | 19.8 |
| | Ciprofloxacin incorrectly identified | 6 | 2.3 |
| **Cause(s) of antimicrobial resistance** | Widespread or overuse of antibiotics | 258 | 98.5 |
| | Usage of broad-spectrum antibiotics | 173 | 66.0 |
| | Genetic mutations of bacteria | 243 | 92.7 |
| | Patient's poor adherence to antibiotic dosage regimens | 240 | 91.6 |
| | Poor hand washing practice in hospitals | 94 | 35.9 |
| | Poor infection control practices in hospital | 142 | 54.2 |
| | Adhering to antibiotic treatment guidelines | 3 | 1.1 |
| | Substandard quality of antibiotics | 193 | 73.7 |

patients will feel their illness is not taken seriously if an antibiotic is not prescribed, while 88 (33.6%) had disagreed and 56 (21.4%) had neither agreed nor disagreed.

When the participants were questioned about their practice of requesting cultures before commencing antibiotics, most who were involved in hospital practice had stated 'almost

**Table 4. Summary of responses for abbreviations used for common antibiotic-resistant bacteria.**

| | Know the meaning n (%) | Don't know the meaning | | Unanswered n (%) |
|---|---|---|---|---|
| | | Heard n (%)* | Not heard n (%)* | |
| **MRSA** | 160 (61.1%) | 100 (99.0%) | 1 (1%) | 1 (0.4%) |
| **CRE** | 38 (14.5%) | 35 (18.2%) | 157 (81.8%) | 32 (12.2%) |
| **ESBL** | 114 (43.5%) | 91 (65%) | 49 (35%) | 8 (3/1%) |
| **VRE** | 69 (26.3%) | 46 (27.1%) | 124 (72.9%) | 23 (8.8%) |

*Percentage calculated from the total number of participants who stated as "Don't know the meaning". MRSA–methicillin resistant *Staphylococcus aureus*, CRE–carbapenem resistant enterobacteriacae, ESBLE–Extended spectrum beta lactamases, VRE–vancomycin resistant enterococcus

**Table 5. Summary of responses for drug of choice for treating resistant bacterial phenotypes.**

| | MRSA | | CRE | | ESBL | | VRE | |
|---|---|---|---|---|---|---|---|---|
| | N | % | n | % | n | % | n | % |
| **Correct choice** | 169 | 64.5 | 32 | 12.2 | 102 | 38.9 | 53 | 20.2 |
| **Correct choice + others** | 47 | 17.9 | 6 | 2.3 | 13 | 5.0 | 3 | 1.1 |
| **Incorrect choice** | 36 | 13.7 | 58 | 22.1 | 74 | 28.2 | 66 | 25.2 |
| **Blank** | 10 | 3.8 | 166 | 63.4 | 73 | 27.9 | 140 | 53.4 |

**Table 6. Perceptions and understanding on antibiotic use.**

|  | Agree | | Disagree | | Neither agree nor disagree | | Unanswered | |
|---|---|---|---|---|---|---|---|---|
|  | n | % | n | % | n | % | n | % |
| I find it hard to select correct antibiotic to a patient with a bacterial infection | 81 | 30.9 | 121 | 46.2 | 59 | 22.5 | 1 | 0.4 |
| Prescribing antibiotics when patient does not need them, doesn't cause any harm | 19 | 7.3 | 234 | 89.3 | 9 | 3.4 | - | - |
| Aware about National Guidelines on antibiotic prescription | 186 | 71.0 | 47 | 17.9 | 28 | 10.7 | 1 | 0.4 |
| Important to know the resistant pattern of the bacteria when selecting an antibiotic | 256 | 97.7 | 3 | 1.1 | 3 | 1.1 | - | - |
| More likely to prescribe antibiotics when the workload is high | 67 | 25.6 | 156 | 59.5 | 37 | 14.4 | 2 | 0.8 |
| Consultation is short when a prescription is issued | 48 | 18.3 | 156 | 59.5 | 54 | 20.6 | 4 | 1.5 |
| Patient will feel their illness is not taken seriously if an antibiotic is not prescribed | 116 | 44.3 | 88 | 33.6 | 56 | 21.4 | 2 | 0.8 |
| Patient education on antibiotics will have an effect on their expectations in later consultation | 243 | 92.7 | 9 | 3.4 | 10 | 3.8 | - | - |
| I feel confident about prescribing antibiotics | 178 | 67.9 | 32 | 12.2 | 50 | 19.1 | 2 | 0.8 |
| I like more education on antibiotic use, resistance and stewardship | 258 | 98.5 | - | - | 3 | 1.1 | 1 | 0.4 |
| I think I practice rational use of antibiotics in my hospital practice | 187 | 71.4 | 18 | 6.9 | 43 | 16.4 | - | - |
| I think I practice rational use of antibiotics in my private practice | 86 | 32.8 | 9 | 3.4 | 28 | 10.7 | 11 | 4.2 |

always or often' (n = 169, 64.5%) while 60 (22.9%) had stated 'sometimes'. Out of those who stated as 'sometimes, rarely or never' (n = 77), 16 had mentioned that no facilities are available while most (n = 26) had mentioned a variety of reasons such as encountering common infections hence not needing culture and their practice is mainly based on clinical assessment. This was different when the same question was analyzed among those who were engaged in private practice (n = 98 responses), in which most participants (n = 47/98) had stated 'sometimes'. Out of those who stated 'sometimes, rarely or never' (n = 68), 18 had stated the high cost as the reason whereas 7 as 'no facilities available', 4 as 'time consuming' and 20 a variety of reasons such as referring the patient to a hospital when needing a culture and not encountering many patients that require cultures. (Table 7).

## Knowledge score

The maximum knowledge score achieved by a participant was 98.31% whereas the minimum was 46.55%. The mean knowledge score was 71.27% with a standard deviation (SD) of 10.83. The median knowledge score was 71.18% and the inter quartile range (IQR) was 64.4%–79.7%. Of all 262 participants, 120 (45.8%) had achieved more than the mean value whereas 257 (98.09%) participants had obtained a knowledge score > = 50%.

## Practice score

The mean practice scores achieved was 65.33% (SD 18.16). The median practice score was 66.67% (IQR of 53.3%– 80%). One-hundred and thirty-seven (52.3%) participants had achieved a score more than the mean value whereas 213 (81.3%) participants had achieved a score >50%.

## Comparison of knowledge and practice scores across groups

The knowledge and practice scores were significantly correlated [Spearman correlation, $p<0.001$, r = 0.343 (Bias corrected 95% CI 0.237–0.448)]. However, the knowledge and practice scores were significantly different ($p<0.001$, Related sample Wilcoxon Signed Rank Test).

**Table 7. Summary of responses for practices regarding antibiotic use.**

| Statement | Response | n | % |
|---|---|---|---|
| **Patients demand antibiotics from you** | Almost always and often | 81 | 30.9 |
| | Sometimes | 135 | 51.5 |
| | Rarely and Never | 43 | 16.4 |
| | Unanswered | 3 | 1.1 |
| **I feel under pressure if my patient expects an antibiotic prescription** | Almost always and often | 43 | 16.4 |
| | Sometimes | 94 | 35.9 |
| | Rarely and never | 122 | 46.6 |
| | Unanswered | 3 | 1.1 |
| **How often do you prescribe antibiotics? (not considered for scoring)** | ≤ once a week | 58 | 22.1 |
| | ≥ 1/day | 113 | 43.1 |
| | 2–4 times/week | 83 | 31.7 |
| | Unanswered | 8 | 3.1 |
| **Do you select antibiotics according to local/ international guidelines?** | Almost always and often | 158 | 60.3 |
| | Sometimes | 72 | 27.5 |
| | Rarely and Never | 27 | 10.3 |
| | Unanswered | 5 | 1.9 |
| **Do you refer BNF (British National Formulary) when prescribing antibiotics?** | Almost always and often | 114 | 43.5 |
| | Sometimes | 111 | 42.4 |
| | Rarely and Never | 36 | 13.7 |
| | Unanswered | 1 | 0.4 |
| **How do you decide the dose of antibiotics (more than one answer accepted)** | By remembering | 173 | 66.0 |
| | By referring to BNF or another formulary | 161 | 61.5 |
| | By following my seniors' practice | 59 | 22.5 |
| | By following my colleagues | 27 | 10.3 |
| **Do you consider AMR when prescribing?** | No | 65 | 24.8 |
| | Yes | 163 | 62.2 |
| | Unanswered | 34 | 13.0 |
| **Do you order cultures before commencing antibiotics in your hospital practice?** | Almost always and often | 169 | 64.5 |
| | Sometimes | 60 | 22.9 |
| | Rarely and never | 17 | 6.4 |
| | Not engaged in hospital practice | 16 | 6.1 |
| **If 'Sometimes', 'Rarely' or 'Never', what is the reason? (not considered for scoring)** | High cost | 8 | |
| | High number of patients | 1 | |
| | No facilities | 16 | |
| | Other | 26 | |
| | Time consuming | 5 | |
| | Unanswered | 21 | |
| **Do you request for cultures before commencing antibiotics in your private practice?*** | Almost always and often | 30 | 30.6* |
| | Sometimes | 47 | 48.0* |
| | Rarely and never | 21 | 21.4* |

(*Continued*)

**Table 7.** (Continued)

| Statement | Response | n | % |
|---|---|---|---|
| | Not engaged in private practice | 151 | |
| | Unanswered | 13 | |
| **If 'Sometimes', 'Rarely' or 'Never', what is the reason? (not considered for scoring)** | High cost | 18 | |
| | No facilities | 7 | |
| | Other | 20 | |
| | Time consuming | 4 | |
| | Unanswered | 19 | |
| **Do you practice de-escalation therapy?** | Yes | 164 | 62.6 |
| | No | 78 | 29.8 |
| | Unanswered | 20 | 7.6 |
| **Do you practice IV to oral switch?** | Yes | 212 | 80.9 |
| | No | 42 | 16.0 |
| | Unanswered | 8 | 3.1 |

*In the demographic section only 83 people acknowledged to engaging in private practice. However, 98 have described their culture related practices in private practice. % calculated with 98 as denominator

Knowledge scores (mean 74.62% and 69.31%, t-test p<0.001) and practice scores (Median 70.59% and 64.71%, Mann Whitney U test p = 0.001) were significantly higher in those with postgraduate qualifications (Table 8).

Table 9 portrays how the participants gather and expand their knowledge on antibiotics. Most (n = 161, 60.8%) had stated that they update themselves on antibiotic prescription by self-study. Other methods being working with seniors (n = 155,58.5%), working with peers (n = 86, 32.5%) and through teaching sessions (n = 109, 41.1%). Most participants had learnt proper antibiotic prescription from consultants (n = 137,52.3%). 63 (24%) had stated that they

**Table 8. Comparison of knowledge and practice scores across different groups.**

| | Sex | | Difference | Involved in or completed post-graduate studies | | Difference | Years in practice | | | Difference |
|---|---|---|---|---|---|---|---|---|---|---|
| | Male (%) | Female (%) | | Yes (%) | No (%) | | 0–5 (%) | 6–10 (%) | >10 (%) | |
| **Mean knowledge score (SD)** | 71.26 (10.87) | 71.27 (10.86) | 0.997 (T-test) | 74.62 (10.33) | 68.39 (10.48) | <0.001 (T-test) | 71.29 (9.85) | 73.77 (12.08) | 68.96 (11.08) | 0.038 (One-Way ANOVA) |
| **Median (IQR)** | 71.19 (64.41–77.97) | 71.19 (64.41–79.67) | | 74.58 (66.10–81.36) | 67.80 (62.71–74.57) | | 71.19 (64.40–79.67) | 72.88 (66.10 --82.20) | 67.80 (61.02–77.97) | |
| **Maximum** | 98.31 | 93.22 | | 98.31 | 91.53 | | 93.22 | 98.31 | 94.92 | |
| **Minimum** | 45.76 | 47.46 | | 47.46 | 45.76 | | 50.85 | 49.15 | 45.76 | |
| **Mean practice score (SD)** | 65.63 (17.72) | 64.88 (18.87) | 0.821 (Mann-Whitney U test) | 69.31 (18.61) | 61.91 (17.11) | 0.001 (Mann-Whitney U test) | 67.61 (15.77) | 65.79 (18.38) | 60.72 (21.22) | 0.062 (Kruskal–Wallis test) |
| **Median (IQR)** | 66.67 (53.85–80.00) | 66.67 (53.33–80.00) | | 70.59 (58.82–82.35) | 64.71 (52.94–73.33) | | 66.67 (60.00–80.00) | 64.71 (53.33–80.00) | 60.18 (46.67–76.47) | |
| **Maximum** | 100 | 100 | | 100 | 93.33 | | 100 | 94.12 | 100 | |
| **Minimum** | 17.65 | 00 | | 00 | 17.65 | | 20 | 26.67 | 00 | |

**Table 9. Expanding knowledge on antibiotics.**

|  |  | n | % |
|---|---|---|---|
| **How do you update yourself on antibiotic prescription on most occasions** | By working with peers | 86 | 32.5 |
|  | By working with seniors | 155 | 58.5 |
|  | Through teaching sessions | 109 | 41.1 |
|  | Self-study | 161 | 60.8 |
| **From whom did you learn the most on proper antibiotic prescription?** | Consultants | 137 | 52.3 |
|  | Peers | 22 | 8.4 |
|  | Self-study | 92 | 35.1 |
|  | Other | 10 | 3.8 |
|  | Unanswered | 1 | 0.4 |
| **What type of training programs would you like, related to antibiotic prescription?** | Workshops | 28 | 10.7 |
|  | Lectures | 49 | 18.7 |
|  | Clinical cases | 16 | 6.1 |
|  | Online course | 63 | 24.0 |
|  | CME | 13 | 5.0 |
|  | Newsletters | 4 | 1.5 |
|  | Unanswered | 71 | 27.1 |

prefer online courses as a useful means to conduct training programmes related to antibiotic prescription while 49 (18.7%) stated as lectures, 28 (10.7%) as workshops, 16(6.1%) as clinical cases, 13(5%) as continuous medical education and 4 (1.5%) as newsletters. 71 (27.1%) had refrained from providing an answer.

## Discussion

This study explores the knowledge and attitude among Sri Lankan doctors with regards to antimicrobial use and resistance. As stated previously, AMR is one of the major threats to global health [1] and the significance of this study lies in the study findings and public health importance in a country where inappropriate antibiotic use is common [10].

Our study highlights relatively higher knowledge scores and comparatively lower practice scores achieved by the participants. A total of 98.09% had achieved a knowledge score more than 50%. The mean value of the percentage score achieved is 71.27% and the median, 71.18%. This is in contrast to the practices score in which 81.3% had achieved a score more than 50%. Here, the mean value was 65.33% and the median, 66.67%.

Most number of participants had 0–5 years of experience hence they are relatively fresh out of undergraduate studies, indicating they may still recall the content from undergraduate curricula while good practices may not be internalized.

Most did not have post graduate qualifications (53.8%) and this indicates that their knowledge updates were from CME (Continuous Medical Education) or if not from the undergraduate studies.

The high knowledge scores exhibited by the study participants is in agreement with another study conducted among junior doctors in primary care centers and hospitals in Crete, Greece [18]. Similar patterns were observed in other studies such as a study conducted among medical doctors in Khyber Pakhtun Khawah, Pakistan [8] and among 2500 Chinese medical and non-medical students [19].

While the overall scores were high, individual questions revealed that some core knowledge on antibiotics, resistance mechanisms and prescription have room for improvement.

Upon asking to select beta lactams, only 35.5% selected at least 1 and only 16.8% selected all four. Some (19.8%) incorrectly picked vancomycin as a beta lactam and only 21.8% knew aztreonam is a beta lactam.

The reason for this result, however, is not clear, but this could have been contributed by the lack of knowledge the participants possess regarding basic classification of beta lactams. Vancomycin is not uncommon in ward practice, and it is a non-beta lactam glycopeptide. Aztreonam on the other hand is not commonly heard of during undergraduate training and since most participants are relatively fresh undergraduates with 0–5 years of experience, the numbers are understandable.

With regards to knowledge in terms associated with common antibiotic-resistant bacteria majority (61.1%) were aware of MRSA. However, the numbers were far less in CRE (14.5%) and VRE (26.3%). These numbers contradict with a cross-sectional survey conducted among Italian young doctors in which 94% participants declared to know what VRE are, 90% to know CRE, 92.9% to know ESBLs and 99% to know MRSA [20]. The difference between these results may be related to their personal experience with patients infected with these multi-drug resistant organisms. Gaps were identified regarding drug choices for treating resistant bacterial phenotypes in which 38.9%, 20.2%, and 12.2% had correctly identified the antibiotics for ESBL, VRE and CRE. This highlights the need for CPD on antimicrobial resistance for updating practice.

A notable number of participants had stated that they find it hard to select the correct antibiotic (30.9%). Majority of our study participants were relatively younger, with 0–5 years experience. These rates are in agreement with previous studies conducted in Greece which identified that younger doctors lacked confidence in antibiotic prescription, particularly in instances where complex decision making is required [18]. Similar findings were present even decades ago where residents have expressed lack of confidence in antibiotic selection [21]. This highlights the need for continuous training, discussions, peer learning and a multi-disciplinary approach to antibiotic use throughout the working career of a medical officer.

The causes for AMR were correctly identified by most participants. However, it is surprising to see that only 35.9% had identified poor handwashing as a contributory factor. Handwashing is one of the most important ways of preventing illness [22] and a cornerstone among efforts to reduce antibiotic resistance [23]. A study reveals that higher knowledge of these factors were positively correlated with having post-graduate training [24].

The knowledge score was significantly associated with post graduate status of the participant and higher knowledge scores were observed among those were either pursuing or had completed their post graduate studies. This outcome is not a surprise since much knowledge and skills are acquired during post graduate training and rational use of antibiotics are learned in depth during clinical training. This knowledge score differed significantly with the duration of medical practice and highest scores were seen among 6–10 years category.

Majority (71%) are aware of national guidelines but only 14.1% would almost always practice selecting antibiotics based on them. These numbers however are not confined to this study. A study conducted in a tertiary care institution at the Caribbean reveals that only 41% and 35% would consult national and institutional guidelines respectively [24]. However, certain studies highlight the importance of treatment guidelines [3, 8] and some demand the need to develop local guidelines [25], a burden Sri Lanka doesn't have to undergo since the Sri Lanka College of Microbiologists in collaboration with other professional colleges and Ministry of Health has published a local guideline already [26].

Majority (97.7%) agreed that knowing the resistant pattern of an organism is beneficial but 64.5% would almost always and often and 22.9% would sometimes order cultures in their hospital practice. Cost and lack of facilities were identified as key reasons for this drawback. Not all hospitals in Sri Lanka are equipped with microbiology laboratory facilities although all cater to a large number of patients on a daily basis. Hence, most practicing doctors in wards would prescribe antibiotics based on their knowledge and past experiences even if a culture is deemed necessary. It is well known that narrowing of antimicrobial spectrum based on culture results is an effective antibiotic management strategy in preventing AMR [27].

A high number of respondents (66%) agreed that they decide the antibiotic dose based on what they remember. In fact, this is the highest preference of majority while 61.5% refer to BNF or another formulary when they need to decide the dose. This preference might pave the way to medical errors since memory is not always accurate and reliable, ultimately leading to poor healing, long hospital stay, high cost and development of AMR at the end of the spectrum. This finding is in line with another study where only 7% of physicians would always take cultures for a suspected infection. The authors had observed that the prohibitive cost and delay in retrieving microbiology reports in some areas have adversely affected perceptions of the value of obtaining routine cultures [24].

Notably, 24.8% of the participants do not consider AMR when prescribing antibiotics, and needless to say this plays a pivotal role in worsening of AMR. This finding maybe due to lack of knowledge and large number of patients who require medical attention, hence limiting the time allocated for an individual patient.

It is commendable to see that majority practice de-escalation therapy (62.6%) and IV to oral switch (80.9%). It is up to the local guidelines to provide precise indications concerning intravenous–oral switch criteria, antibiotic combination choice criteria, and optimal durations of antibiotic treatments [3].

The practices score was also significantly associated with the postgraduate status of the participant just as the knowledge scores. This outcome maybe contributed by reasons stated previously which are common in both instances.

The knowledge and practice scores correlated significantly. This implies that possessing high and up-to-date knowledge in antibiotic use has a significant impact on correct antibiotic practices.

Our study has certain limitations. The practice was assessed using a questionnaire than using direct observations. However, we believe that the anonymous nature of the questionnaire would have made more people to answer the questions truthfully. Further, even though we validated and piloted the tool, we did not calculate the reliability, validity or other measures statistically.

## Conclusions and recommendations

This study concludes that the practitioners generally had a good level of knowledge regarding antibiotic use. However, there is room for improvement in knowledge regarding beta lactams and selection of antibiotics for different resistant phenotypes.

Comparatively lower practices scores regarding antibiotic use were noted. The knowledge and practices of a doctor are positively correlated with the post graduate status highlighting the fact that knowledge and practices are refined during post graduate training. The knowledge score was also associated with duration of clinical practice.

This study also reflects the need to regulate the prescription of antibiotics and initiate training programmes for doctors in order to improve their practices. This also highlights potential causes for AMR hence aforementioned training programmes should focus more on behavior

rather than knowledge alone. Furthermore, doctors must be encouraged to use local guidelines and CME is an essential step in promoting rational use of antibiotics, despite Sri Lanka being a country where CME is not mandatory for license renewal.

## Supporting information

**S1 Table. Mark allocation for knowledge scores.**
(DOCX)

**S2 Table. Mark allocation for practice scores.**
(DOCX)

**S1 Database.**
(XLSX)

## Author Contributions

**Conceptualization:** Kaushika Jayawardena, Asanka Tennegedara, Veranja Liyanapathirana.

**Data curation:** Gihan Shu, Kaushika Jayawardena, Dinesh Jayaweera Patabandige, Veranja Liyanapathirana.

**Formal analysis:** Gihan Shu, Veranja Liyanapathirana.

**Investigation:** Dinesh Jayaweera Patabandige, Veranja Liyanapathirana.

**Methodology:** Kaushika Jayawardena, Dinesh Jayaweera Patabandige, Asanka Tennegedara, Veranja Liyanapathirana.

**Project administration:** Dinesh Jayaweera Patabandige, Veranja Liyanapathirana.

**Supervision:** Asanka Tennegedara, Veranja Liyanapathirana.

**Writing – original draft:** Gihan Shu, Veranja Liyanapathirana.

**Writing – review & editing:** Gihan Shu, Kaushika Jayawardena, Dinesh Jayaweera Patabandige, Asanka Tennegedara, Veranja Liyanapathirana.

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
