## [Decision Letter · Decision Letter 0]

29 Nov 2021

PONE-D-21-34680Knowledge and practices on antibiotic use among Sri Lankan doctorsPLOS ONE

Dear Dr. Liyanapathirana,

Thank you for submitting your manuscript to PLOS ONE. After careful consideration, we feel that it has merit but does not fully meet PLOS ONE’s publication criteria as it currently stands. Therefore, we invite you to submit a revised version of the manuscript that addresses the points raised during the review process.

Kindly go through the reviewrers' comments and the attached document carefully and respond to the comments and modify the submitted docuemnt. 

We look forward to receiving your revised manuscript.

Kind regards,

Pathiyil Ravi Shankar

Academic Editor

PLOS ONE

Journal Requirements:

a) Did participants provide their written or verbal informed consent to participate in this study?

3. Thank you for stating the following in the Competing Interests/Financial Disclosure * (delete as necessary) section: 

(VL has received funding from Pfizer for a non-related study. Others declare no competing interests)

We note that you received funding from a commercial source: (Pfizer)

Reviewers' comments:

Reviewer's Responses to Questions

**Comments to the Author**

1. Is the manuscript technically sound, and do the data support the conclusions?

Reviewer #1: Partly

Reviewer #2: Yes

Reviewer #3: Yes

2. Has the statistical analysis been performed appropriately and rigorously? 

Reviewer #1: I Don't Know

Reviewer #2: Yes

Reviewer #3: I Don't Know

3. Have the authors made all data underlying the findings in their manuscript fully available?

Reviewer #1: Yes

Reviewer #2: No

Reviewer #3: Yes

4. Is the manuscript presented in an intelligible fashion and written in standard English?

Reviewer #1: No

Reviewer #2: Yes

Reviewer #3: Yes

5. Review Comments to the Author

Reviewer #1: PONE-D-21-34680

Research Article

Knowledge and practices on antibiotic use among Sri Lankan doctors

Reviewer’s Comment

General Comments: The manuscript requires copyediting for both content and English language.

Specific Comments:

Abstract:

Introduction: It would be better to avoid using abbreviations in abstract section. If it is used, follow only the internationally adopted abbreviations. I also suggest the author to use the term ‘antimicrobial’ rather than antibiotic.

Methods: The information provided in method section lack vital information about how the questionnaire was developed? Who were the study participants? How questionnaire was applied? How validation of the questionnaire was performed? How the scoring was executed? How validated response can be obtained through google form? Likert scale questions are generally used to assess the attitude of the respondents, but here it has been used for knowledge/practice?

Key words: The authors should make sure the keywords are MeSH words, particularly knowledge and practices,

Main body:

Introduction: Reorganize this section in different paragraph with background information, problem statement, rationale of the study

Methodology: It will be worthwhile mentioning the qualification and experiences of the experts who were involved in validation of the questionnaire. How reliability of the questionnaire was performed? Why different scoring was used, e.g. 0.25 score, 1 score, negative score, 0 score, 2 score? These are too confusing! The term ‘mark/s’ should better be replaced with the term ‘score/s’.

The information provided in method section lack vital information about how the questionnaire was developed? Who were the study participants? How questionnaire was applied? One respondent can fill the google questionnaire multiple times. How validated response can be obtained through google form? Likert scale questions are generally used to assess the attitude of the respondents, but here it has been used for knowledge/practice? Duration and location of the study?

Results: the number of unanswered responses are seen in few cases only, it would be better to merge them as ‘Don’t know/Unanswered’(table 2). Why the heading for Table 4 is below the table? Full form of the abbreviations used in table 4 should be mentioned in footer of the table. Table 6 is about the perception on antibiotic use, whereas there is no any word about perception in the title of the manuscript? It is quite unusual to use 4 points likert scale (table 6). It can be made into 3 scales (agree, Neutral/unanswered, disagree).

Practice can better be determined through observation of real prescriptions (table 7) not through asking the questions. The responses for this section through questionnaire particularly online will be biases-how many will say that they do malpractice???

It would be meaningful to use the median score rather than mean score.

There are nine tables in this manuscript. It is advisable to reduce the number of tables if feasible through merging or modifying the tables.

Discussion: Discussion require more comparisons and clarifications. Study limitations and recommendations are vital components of a manuscript.

Conclusions: This section should be short and conclusive with key message from the findings.

References: Make sure the references and their citations in the text are as per the journal’s requirements.

Reviewer #2: 1. Title - did the authors also study the aspect related to resistance?

2. Abstract - ABR or AMR?

3. The objective in the abstract, main text as well as findings and conclusions should be consistent; look at the ms very carefully - the terms use, resistance, prescription etc were used inconsistently

4. Abstract - methods - too brief, how was the data analyzed?

5. This topic has been studied in the past and many articles have been published. What the study adds to the literature, practice and policy? What is new?

6. main text - obj - I thought the study also look at knowledge and practice related ABc resistance!

7. Methods - was poorly written - non structured and non systematic way of writing the methods section - difficult for readers to follow and not easy to be replicated; what was the sample size and power of the study? How the respondents were selected? the main domain - what is the possible min and max values/scores for practice? what are the possible min and max scores for knowledge?; Was any psychometric measures done on the tool? Was the study piloted? alpha value? what software was used for the analysis? mean values should be followed by sd, and median with IQR

8. any limitations of the study? what are the study implications? any recommendations?

Reviewer #3: Authors have done a good work. There are some comments and suggestions in the manuscript submitted. Please go through the comments and suggestions and do the necessary corrections for a revised submission.

6. PLOS authors have the option to publish the peer review history of their article (what does this mean?). If published, this will include your full peer review and any attached files.

Reviewer #1: **Yes: **Mukhtar Ansari

Reviewer #2: **Yes: **Mohamed Izham Mohamed Ibrahim

Reviewer #3: No

---

## [Author Response · Author response to Decision Letter 0]

21 Dec 2021

PONE-D-21-34680

Knowledge and practices on antibiotic use among Sri Lankan doctors

PLOS ONE

1. Formatting has been done according to the two style guides provided 

a) Did participants provide their written or verbal informed consent to participate in this study?

The questionnaire contained the following segment at the start of the questionnaire. Ethical clearance application stated that informed consent would be taken by participants through online means. The explicit way of doing was not included in the ethics application. 

Relevant section from ethics application 

“An informed consent will be obtained at the start of the survey. All information regarding the study, participants’ rights and researcher’s contact details are provided on the first page of the online survey link directed via mobile application-WhatsApp/Viber. Information on personal identity such as name, address, contact details and citizenship ID number, SLMC reg number shall not be obtained in the questionnaire.

Participants are assured that their responses are treated anonymously and no individual will be identified to maintain the confidentiality of the information. The collected data and records shall be held by the members of the research team, made available only to the facilitator, ethical committee and other relevant departments and will not be made accessible to the public. Only the final report will be published.

Any participant has the right to continue or to refuse answering the questionnaire. Only the subjects who have given consent will be proceeded to the survey.”

The first section of the online survey 

“This questionnaire is part of a study to assess knowledge, attitudes and practices on antibiotic use and resistance among medical doctors in Sri Lanka with the aim of developing targeted educational programmes. 

This study is being conducted by Dr.Kaushika Jayawardene, Dr.Veranja Liyanapathirana and Dr.Asanka Tennegedara from Department of Microbiology, Faculty of Medicine, University of Peradeniya. 

Ethical clearance was obtained from the Ethics Review Committee, Faculty of Medicine, University of Peradeniya under the Project No.2020/EC/36.

We invite you to kindly complete this questionnaire by clicking the appropriate boxes.

We invite you to answer this questionnaire individually, without discussing or referring to content.

Your responses, will be treated as confidential and analyzed data will be used for the sole purpose of scientific communications and content development.

Further information on the study is available from https://drive.google.com/file/d/1ALCJrQkzIJwLdF-mPacLkm5zXxtYnkGS/view?usp=sharing

Your clicking the "next" button here will be taken as giving consent to participate in this study. 

THANK YOU for choosing to fill the questionnaire independently without any help.” 

As this was clearly stated at the start of the questionnaire and the participants had access to the information sheet, along with the fact that no personal identifiers were collected or analyzed and that participants were medical officers who are sufficiently competent to comprehend the introductory section of the questionnaire, authors feel that this information and way of obtaining consent is sufficient. 

3. Thank you for stating the following in the Competing Interests/Financial Disclosure * (delete as necessary) section: 

(VL has received funding from Pfizer for a non-related study. Others declare no competing interests)

We note that you received funding from a commercial source: (Pfizer)

The amended competing interests statement included within the cover letter reads as “VL has received funding from Pfizer for a non-related study. Others declare no competing interests. "This does not alter our adherence to PLOS ONE policies on sharing data and materials.”

I would also like to emphasize that this study received not funding from anywhere. 

The two supplementary tables were renamed. And the list was included at the end of the manuscript before the references. 

Reviewers' comments:

Comments to the Author

1. Is the manuscript technically sound, and do the data support the conclusions?

Reviewer #1: Partly

Reviewer #2: Yes

Reviewer #3: Yes

2. Has the statistical analysis been performed appropriately and rigorously?

Reviewer #1: I Don't Know

Reviewer #2: Yes

Reviewer #3: I Don't Know

3. Have the authors made all data underlying the findings in their manuscript fully available?

Reviewer #1: Yes

Reviewer #2: No – Authors would like to inform that the completed database has been uploaded as a supplementary material which the reviewer may have missed 

Reviewer #3: Yes

4. Is the manuscript presented in an intelligible fashion and written in standard English?

Reviewer #1: No

Reviewer #2: Yes

Reviewer #3: Yes

5. Review Comments to the Author

Reviewer #1: PONE-D-21-34680

Research Article

Knowledge and practices on antibiotic use among Sri Lankan doctors

Reviewer’s Comment

General Comments: The manuscript requires copyediting for both content and English language.

Thank you, we have gone through and improved the manuscript where possible. 

Specific Comments:

Abstract:

Introduction: It would be better to avoid using abbreviations in abstract section. If it is used, follow only the internationally adopted abbreviations. I also suggest the author to use the term ‘antimicrobial’ rather than antibiotic.

Thank you. We have replaced antibiotic resistance with antimicrobial resistance and used the abbreviation AMR, which is widely known now. 

Methods: The information provided in method section lack vital information about how the questionnaire was developed? Who were the study participants? How questionnaire was applied? How validation of the questionnaire was performed? How the scoring was executed? How validated response can be obtained through google form? Likert scale questions are generally used to assess the attitude of the respondents, but here it has been used for knowledge/practice?

Thank you. We have included the sentence, “The Google sheet generated was used for data analysis”. Given the restriction on the number of wording, authors are unable to include all requested details in the methods section within the abstract. Details on how the marking was done is given in the methods section in detail along with detailed mark break-down in the supplementary material. Participants are indicated to in the objectives, therefore a repetition was not done in the methods section. 

Key words: The authors should make sure the keywords are MeSH words, particularly knowledge and practices,

Thank you. The keywords were updated to 

Health Knowledge, Attitudes, Practice

Antibiotics 

Main body:

Introduction: Reorganize this section in different paragraph with background information, problem statement, rationale of the study

Authors feel that the current organization of the introduction includes background information, a problem statement and the rationale therefore no change was done to the introduction. 

Methodology: It will be worthwhile mentioning the qualification and experiences of the experts who were involved in validation of the questionnaire. 

The methods section already mentioned that the questionnaire was validated by a medical educationist and two consultant microbiologists. Now it reads as “It was validated by a Specialist Medical Educationist and two Consultant Microbiologists with experience in similar studies and relevant clinical practice respectively.”

How reliability of the questionnaire was performed? Why different scoring was used, e.g. 0.25 score, 1 score, negative score, 0 score, 2 score? These are too confusing! The term ‘mark/s’ should better be replaced with the term ‘score/s’.

Thank you for pointing this out. We accept that the assignment of marks may be a bit confusing on first reading. However, the supplementary tables indicates the allocation in a clear way. These were decided up on by all researchers. We have replaced the mark/s with score/scores where appropriate but retained the word mark where it is more appropriate. 

The information provided in method section lack vital information about how the questionnaire was developed? Who were the study participants? How questionnaire was applied? One respondent can fill the google questionnaire multiple times. How validated response can be obtained through google form? Duration and location of the study?

Thank you. The methods section now reads – “This was a cross-sectional study conducted using recruiting 262 participants from various fields of Medicine working in different units in Sri Lanka. The ethical clearance was obtained from the Ethics Review Committee, Faculty of Medicine, University of Peradeniya, Sri Lanka (2020/EC/36) and participants indicated consent for participation through pressing the next button as informed through instructions. The questionnaire was Google form-based. It was validated by a Specialist Medical Educationist and two Consultant Microbiologists with experience in similar studies and relevant clinical practice respectively. It was disseminated to doctors through hospital based social media groups. All doctors practicing in Sri Lanka, registered with the Sri Lanka Medical Council were eligible to answer the questions. One attempt was allowed per-email address; therefore a single response was taken per-participant. The questionnaire was open to accept answers from 15/10/2021 to 12/01/2021”. We hope this gives sufficient information. 

Likert scale questions are generally used to assess the attitude of the respondents, but here it has been used for knowledge/practice? 

Thank you. While Likert scales are usually used for attitudes, they are been used to measure knowledge and practices previously too. We have converted the marks and the methods of conversion are also given. 

Results: the number of unanswered responses are seen in few cases only, it would be better to merge them as ‘Don’t know/Unanswered’(table 2). 

Thank you for the suggestion. As for the rest of the analysis, the unanswered segments vary, and it is important to identify them, we prefer to keep the unanswered segment separately in Table 2 as well. 

Why the heading for Table 4 is below the table? Full form of the abbreviations used in table 4 should be mentioned in footer of the table. 

The title of table 4 is placed above it. It may be due to a technical error in generating the pdf that reviewer has seen it such. The footnote defining abbreviations were added. 

Table 6 is about the perception on antibiotic use, whereas there is no any word about perception in the title of the manuscript? It is quite unusual to use 4 points likert scale (table 6). It can be made into 3 scales (agree, Neutral/unanswered, disagree).

Thank you very much. The title was changed to include perceptions. We feel that it is important to highlight the non-response rate, therefore would like to keep the two columns separately, unless the editors would like the two columns to be combined as well. 

Practice can better be determined through observation of real prescriptions (table 7) not through asking the questions. The responses for this section through questionnaire particularly online will be biases-how many will say that they do malpractice???

Thank you for pointing this out. However, even directly observed studies will have this bias. We believe that the fact that participants have acknowledged some malpractices and the anonymous nature of the questionnaire will improve correct responses. However, we have now included this as a shortcoming of the study. 

It would be meaningful to use the median score rather than mean score.

We agree, therefore we have given both the mean and the median scores in all places. 

There are nine tables in this manuscript. It is advisable to reduce the number of tables if feasible through merging or modifying the tables.

Thank you. We have limited the tables as much as possible to get the nine tables. Editorial recommendations on which further tables to merge would be most welcome. 

Discussion: Discussion require more comparisons and clarifications. Study limitations and recommendations are vital components of a manuscript.

Thank you. We have now included a limitations section and the recommendations were already included in the conclusion section. 

Conclusions: This section should be short and conclusive with key message from the findings.

Thank you for pointing this out. We believe that it contains the main messages only and includes the recommendations and have renamed the section as conclusions and recommendations. If the editors wishes us to do, we can modify the section and remove the recommendations. 

References: Make sure the references and their citations in the text are as per the journal’s requirements.

Reviewer #2: 

1. Title - did the authors also study the aspect related to resistance?

We did not ask about resistance from the point of view of discussing about resistance. So would like to keep it as antibiotics. However, in response to reviewer 1, we have added “perceptions” to the title 

2. Abstract - ABR or AMR?

Thank you. ABR has been replaced with AMR

3. The objective in the abstract, main text as well as findings and conclusions should be consistent; look at them very carefully - the terms use, resistance, prescription etc were used inconsistently

Thank you for pointing out, we have gone through the manuscript again and modified where we felt it was needed. 

4. Abstract - methods - too brief, how was the data analyzed?

We expanded the methods section to include administration of the questionnaire and some other aspects. We feel that the data analysis is already sufficiently described. However, if the editor wishes so, we would be able to expand it a bit more. 

5. This topic has been studied in the past and many articles have been published. What the study adds to the literature, practice and policy? What is new?

6. main text - obj - I thought the study also look at knowledge and practice related ABc resistance!

We look at the antibiotic use, in the context of AMR. This was added to the text. The main focus is actually not AMR but the current level of KAP on antibiotics in the context of AMR.

7. Methods - was poorly written - non structured and non systematic way of writing the methods section - difficult for readers to follow and not easy to be replicated; 

Thank you, we have revised the methods section to be more systematic. If the editors wishes us to do so, we are agreeable to move the description on the scoring system to the supplementary material, along with the two tables. 

what was the sample size and power of the study?

The sample size was calculated to be 384, but we analyzed the received samples, and we have now stated that convenience sampling was used. It is globally knowns that physicians are a very difficult group to do surveys on, therefore, authors feel that the sample size available is adequate.

Cho YI, Johnson TP, Vangeest JB. Enhancing surveys of health care professionals: a meta-analysis of techniques to improve response. Eval Health Prof. 2013 Sep;36(3):382-407. doi: 10.1177/0163278713496425. PMID: 23975761.

Taylor T, Scott A. Do Physicians Prefer to Complete Online or Mail Surveys? Findings From a National Longitudinal Survey. Eval Health Prof. 2019 Mar;42(1):41-70. doi: 10.1177/0163278718807744. Epub 2018 Nov 1. PMID: 30384770. 

How the respondents were selected? 

As stated above, and in the methods section, anyone who was lisenced to practice medicine in Sri Lanka were eligible to participate. This is indicated in the method section. 

the main domain - what is the possible min and max values/scores for practice? what are the possible min and max scores for knowledge?; 

Thank you for the comment. The following paragraph has been added to the methods section “The maximum mark achievable for knowledge score was 14.75 while it differed for the practice score depending on if participants were engaged in hospital practice and/or private practice. All total marks were converted to %, so the maximum achievable mark was 100% while the minimum was 0%.” 

Further, this was already presented in the results section but we deleted it after including in the methods section. 

Was any psychometric measures done on the tool? Was the study piloted? alpha value? 

The tool was piloted, it is mentioned in the methods section. No psychometric analysis was done. We did not measure the alpha value. The use of alpha value for validation of questionnaires is being debated. As our study population is relatively homogenous, authors feel that calculation of alpha value is not needed. 

Taber, K.S. The Use of Cronbach’s Alpha When Developing and Reporting Research Instruments in Science Education. Res Sci Educ 48, 1273–1296 (2018). https://doi.org/10.1007/s11165-016-9602-2

What software was used for the analysis? 

Thank you, this was included in the methods section now. SPSS version 23 was used for data analysis.

Mean values should be followed by sd, and median with IQR

Mean values were given with the SD, it was added to where it was not given, IQR was added to the median

8. any limitations of the study? what are the study implications? any recommendations?

Limitations section was added while the conclusion section was changed to conclusions and recommendations 

Reviewer #3: Authors have done a good work. There are some comments and suggestions in the manuscript submitted. Please go through the comments and suggestions and do the necessary corrections for a revised submission.

Thank you. However, the manuscript with comments was not available either as an attachment, within the email or in the journal’s submission portal.

6. PLOS authors have the option to publish the peer review history of their article (what does this mean?). If published, this will include your full peer review and any attached files.

Do you want your identity to be public for this peer review? For information about this choice, including consent withdrawal, please see our Privacy Policy.

Reviewer #1: Yes: Mukhtar Ansari

Reviewer #2: Yes: Mohamed Izham Mohamed Ibrahim

Reviewer #3: No

---

## [Decision Letter · Decision Letter 1]

27 Dec 2021

PONE-D-21-34680R1Knowledge, perceptions and practices on antibiotic use among Sri Lankan doctorsPLOS ONE

Dear Dr. Liyanapathirana,

Thank you for submitting your manuscript to PLOS ONE. After careful consideration, we feel that it has merit but does not fully meet PLOS ONE’s publication criteria as it currently stands. Therefore, we invite you to submit a revised version of the manuscript that addresses the points raised during the review process.

Kindly address all the commets of the reviewer and kindly copyedit your manuscript by a native Englsih speaker or an editring service. 

We look forward to receiving your revised manuscript.

Kind regards,

Pathiyil Ravi Shankar

Academic Editor

PLOS ONE

Reviewers' comments:

Reviewer's Responses to Questions

**Comments to the Author**

1. If the authors have adequately addressed your comments raised in a previous round of review and you feel that this manuscript is now acceptable for publication, you may indicate that here to bypass the “Comments to the Author” section, enter your conflict of interest statement in the “Confidential to Editor” section, and submit your "Accept" recommendation.

Reviewer #1: (No Response)

Reviewer #3: All comments have been addressed

2. Is the manuscript technically sound, and do the data support the conclusions?

Reviewer #1: Yes

Reviewer #3: Yes

3. Has the statistical analysis been performed appropriately and rigorously? 

Reviewer #1: I Don't Know

Reviewer #3: I Don't Know

4. Have the authors made all data underlying the findings in their manuscript fully available?

Reviewer #1: Yes

Reviewer #3: Yes

5. Is the manuscript presented in an intelligible fashion and written in standard English?

Reviewer #1: No

Reviewer #3: Yes

6. Review Comments to the Author

Reviewer #1: Authors are advised to address all of the comments. The write up of the manuscript still requires language editing.

Reviewer #3: The authors have answered to all the queries made by the reviewers. The manuscript now in a better form and ready to be accepted.

7. PLOS authors have the option to publish the peer review history of their article (what does this mean?). If published, this will include your full peer review and any attached files.

Reviewer #1: **Yes: **Mukhtar Ansari

Reviewer #3: No

---

## [Author Response · Author response to Decision Letter 1]

3 Jan 2022

Author response to reviewer comments 

Comments to the Author

Reviewer #1: No

Author response: We have corrected the manuscript and hope the language it clearer now 

Reviewer #3: Yes

6. Review Comments to the Author

Reviewer #1: Authors are advised to address all of the comments. The write up of the manuscript still requires language editing.

Author response: We could not find any reviewer comments to address. We had given our responses to the reviewer comments in our previous revisions, we would be most grateful if the reviewer can specify which comments need further addressing. 

Thank you for suggesting language edits, we have attended to it. We hope the reviewer and the editors are satisfied with the changes made and look forward to any specific comments that may still need corrections. 

Reviewer #3: The authors have answered to all the queries made by the reviewers. The manuscript now in a better form and ready to be accepted.

---

## [Decision Letter · Decision Letter 2]

13 Jan 2022

Knowledge, perceptions and practices on antibiotic use among Sri Lankan doctors

PONE-D-21-34680R2

Dear Dr. Liyanapathirana,

We’re pleased to inform you that your manuscript has been judged scientifically suitable for publication and will be formally accepted for publication once it meets all outstanding technical requirements.

Kind regards,

Pathiyil Ravi Shankar

Academic Editor

PLOS ONE

Additional Editor Comments (optional):

Reviewers' comments:

Reviewer's Responses to Questions

**Comments to the Author**

1. If the authors have adequately addressed your comments raised in a previous round of review and you feel that this manuscript is now acceptable for publication, you may indicate that here to bypass the “Comments to the Author” section, enter your conflict of interest statement in the “Confidential to Editor” section, and submit your "Accept" recommendation.

Reviewer #1: All comments have been addressed

2. Is the manuscript technically sound, and do the data support the conclusions?

Reviewer #1: Yes

3. Has the statistical analysis been performed appropriately and rigorously? 

Reviewer #1: I Don't Know

4. Have the authors made all data underlying the findings in their manuscript fully available?

Reviewer #1: Yes

5. Is the manuscript presented in an intelligible fashion and written in standard English?

Reviewer #1: Yes

6. Review Comments to the Author

Reviewer #1: The authors have addressed all of the comments, and the manuscript looks fine after the revisions have been made.

7. PLOS authors have the option to publish the peer review history of their article (what does this mean?). If published, this will include your full peer review and any attached files.

Reviewer #1: **Yes: **Mukhtar Ansari

---

## [Editor Report · Acceptance letter]

19 Jan 2022

PONE-D-21-34680R2 

Knowledge, perceptions and practices on antibiotic use among Sri Lankan doctors 

Dear Dr. Liyanapathirana:

I'm pleased to inform you that your manuscript has been deemed suitable for publication in PLOS ONE. Congratulations! Your manuscript is now with our production department. 

Kind regards, 

on behalf of

Dr. Pathiyil Ravi Shankar 

Academic Editor

PLOS ONE